# Real-Time Water Surface Object Detection Based on Improved Faster R-CNN

**DOI:** 10.3390/s19163523

**Published:** 2019-08-12

**Authors:** Lili Zhang, Yi Zhang, Zhen Zhang, Jie Shen, Huibin Wang

**Affiliations:** College of Computer and Information Engineering, Hohai University, Nanjing 211100, China

**Keywords:** object detection, deep learning, real-time, Faster R-CNN

## Abstract

In this paper, we consider water surface object detection in natural scenes. Generally, background subtraction and image segmentation are the classical object detection methods. The former is highly susceptible to variable scenes, so its accuracy will be greatly reduced when detecting water surface objects due to the changing of the sunlight and waves. The latter is more sensitive to the selection of object features, which will lead to poor generalization as a result, so it cannot be applied widely. Consequently, methods based on deep learning have recently been proposed. The River Chief System has been implemented in China recently, and one of the important requirements is to detect and deal with the water surface floats in a timely fashion. In response to this case, we propose a real-time water surface object detection method in this paper which is based on the Faster R-CNN. The proposed network model includes two modules and integrates low-level features with high-level features to improve detection accuracy. Moreover, we propose to set the different scales and aspect ratios of anchors by analyzing the distribution of object scales in our dataset, so our method has good robustness and high detection accuracy for multi-scale objects in complex natural scenes. We utilized the proposed method to detect the floats on the water surface via a three-day video surveillance stream of the North Canal in Beijing, and validated its performance. The experiments show that the mean average precision (MAP) of the proposed method was 83.7%, and the detection speed was 13 frames per second. Therefore, our method can be applied in complex natural scenes and mostly meets the requirements of accuracy and speed of water surface object detection online.

## 1. Introduction

In recent years, object detection technology has attracted a great deal of attention within the computer vision community [1,2,3,4,5], as an important component of many human-centric applications. Generally, object detection methods include background subtraction [6], frame difference [7], Hough transform [8], optical flow [9,10] and so on. These methods require hand-crafted models to extract specific features, and the extracted features have deficiencies in representativeness and robustness, which leads to poor generalization ability as a result. Therefore, object detection methods based on deep learning have emerged and been successfully applied in self-driving vehicles [1], face recognition [2], and pedestrian detection [11]. The methods based on deep learning models have stronger expressive ability than the classical methods.

The problem of water environmental pollution is serious in some countries. Floats such as plastic bottles and bags affect water quality and the urban living environment, and a large number of floats can have an impact on the passage of ships [12]. Hence, water surface floats, especially garbage, should be detected and cleaned in good time. In China, the government began to focus on this, and established the River Chief System, which requires the detection of water surface floats in a timely fashion. The administration usually monitors the river situation, relying on video surveillance or on-the-spot inspections by inspectors and cleaners [13,14,15]. This is time-consuming and not intelligent. Many classical works [16,17,18,19] are devoted to water surface object detection and can be classified into background subtraction and image segmentation. The former is highly susceptible to the variance of the scenes, so its accuracy is greatly reduced when detecting water surface objects due to the variance of the sunlight and waves. The latter is more sensitive to the selection of object features, which leads to poor generalization as a result, so it cannot be applied widely. In the case of complex scenes, the accuracy and robustness are greatly reduced. Faster R-CNN has strong generalization ability and high detection accuracy in some fields. Therefore, we adopted this network model as the key element for water surface object detection. 

Our proposed method attempts to preprocess the negative effect of sunlight firstly via gamma correction [20], and we propose a deep learning network model based on Faster R-CNN, in which we integrate high-level features with low-level features in order to improve the detection accuracy. There are different receptive fields in different layers of a CNN, and the larger receptive fields often contain a great deal of useless background noises for small-scale objects. Therefore, we selected the proper convolutional layers for feature fusion based on the receptive field. In addition, we did not use the uniform standard to set the scales, aspect ratios, and numbers of anchors in Faster R-CNN. In this paper, we propose to set the scales, aspect ratios, and numbers of anchors by analyzing the size distribution of water surface objects in our dataset, solving the object detection of different scales. Experiments on our collected dataset verified that the proposed method in this paper could achieve an excellent performance in multi-scale water surface object detection in complex scenes.

## 2. Related Work

The classical methods for water surface object detection mainly include background subtraction and image segmentation. Jie Jiang et al. [16] proposed background subtraction and frame subtraction based on mixture Gauss model to detect water surface floats, which solved the problem of manually monitoring floats that the buoy requires. Youfu Wu et al. [17] proposed the adaptive Gaussian mixture model to model a complex background, which improved the accuracy of float detection. These methods can achieve high detection accuracy in static background scenes. However, they are susceptible to background such as the change of sunlight, rain, and surface fluctuation, which leads to poor performance. Jianjun Zuo [18] segmented the floats by background difference, extracted object color features, and finally classified and recognized the floats by a clustering algorithm. Rong Hu [19] adopted the edge segmentation method and threshold segmentation method based on maximum entropy to segment water surface objects. They compared the efficiency of BP (back propagation) neural network, KNN (k-nearest neighbor) algorithm, and SVM (support vector machine) on image classification and recognition. Finally, it was verified that the KNN algorithm had the highest recognition accuracy in many applications. These methods are more sensitive to the selection of object features, which leads to poor generalization ability. In addition, the high computing costs of these methods make real-time performance a bottleneck. 

Consequently, AlexNET [21] has been successfully applied to image recognition and made great breakthroughs in image classification tasks. With the extensive applications of deep learning in computer vision, object detection based on deep learning has also made significant breakthroughs. Ross Girshick et al. [22] developed the region-based convolutional neural network (R-CNN) object detection algorithm to improve the accuracy. However, R-CNN performs convolution operations for each object proposal without sharing computation, so it has a high cost. Subsequently, Ross Girshick [23] proposed Fast R-CNN—a fast update based on R-CNN and SPPnet [24]. Compared to R-CNN, Fast R-CNN improves the training and testing speed while increasing detection accuracy. However, Fast R-CNN still retains the selective search strategy [25], making region proposal computation a bottleneck. To solve this problem, Shaoqing Ren et al. [4] proposed Faster R-CNN, which introduced a region proposal network (RPN) that shares the extracted features with the detection network, thus achieving nearly cost-free generation of region proposals. The RPN also improves the quality of region proposals. The method enables object detection systems based on deep learning to run at near-real-time frame rates. In order to further improve the detection speed, Joseph Redmon [26] proposed the YOLO algorithm, which frames the object detection as a regression problem to improve the detection speed. The advantage of this method is that it can detect dozens of frames per second, but the detection for small objects is insensitive, which reduces the detection accuracy. 

Recently, many studies have improved object detection by using multi-scale representation. In [27,28], skip-layer connection is adopted to extract features. The information gained is especially important for small objects. The results suggest that multi-scale representation can improve small-object detection. Tsung-Yi Lin et al. [29] proposed feature pyramids based on Faster R-CNN. The strong semantic and low-resolution features are fused with the high-resolution and weak semantic features by a top-down pathway and lateral connections, thus building high-level semantic feature maps at all scales. Jianan Li et al. [30] combined the global and local contextual information into region proposal for object detection, which achieved better object detection performance.

## 3. Water Surface Object Detection Method

Due to the excellent performance of Faster R-CNN in object detection, we made it the key part in our method to detect water surface objects. Our method can be summarized as follows:

(1) *Image preprocessing*: Variable sunlight leads to over-exposed or under-exposed images, which makes it difficult to distinguish floats from the background. A gamma correction algorithm is adopted to enhance the contrast of images.

(2) *Constructing a network model*: Our network model is an improved Faster R-CNN. Feature fusion is utilized for feature extraction in order to improve the final detection accuracy. Besides, the different scales, aspect ratios, and numbers of anchors in RPN are set by analyzing the distribution.

(3) *Training of network:* We adopt the alternating optimization method to train our network model. 

(4) *Detection*: The trained network is used to detect the water surface objects and thus validate the performance of our network model and our method. 

### 3.1. Image Preprocessing

The variable sunlight in natural scenes will lead to over-exposed or under-exposed images. The floats and background are difficult to distinguish, so the shape features of the floats are not clear, which will affect the float detection. Therefore, we utilize the gamma correction algorithm to enhance the contrast of images, while maintaining the integrity of image information.

Gamma correction is a non-linear operation used to encode and decode luminance or tristimulus values in videos or images. Gamma correction is defined by the following formula:(1)$$V_{out}=AV_{in}^{\gamma },$$
where the real input value $V_{in}$ is raised to the power γ and multiplied by the constant *A*, to get the output value $V_{out}$. In general, *A* = 1. 

When the gamma value is greater than 1, the highlighted parts of the image are suppressed, and the dark parts are simultaneously expanded. Conversely, when the gamma value is less than 1, the highlighted parts of the image are be expanded and the dark parts are suppressed. It is difficult to distinguish the floats from the background in the images, so the dark parts of the image need to be expanded, and we set the gamma value to be less than 1. We tried different gamma values for experiments, and the image was enhanced the best when the gamma value was 0.6. The image contrasts produced by different gamma corrections are shown in Table 1, and the corrected images are shown in Figure 1.

### 3.2. Scale-Aware Network Model Based on Faster R-CNN

Our proposed network model is composed of two modules. One is a deep fully convolutional network called an RPN (region proposal network), and the other is the Fast R-CNN detector. Figure 2 shows our network model. Firstly, image feature maps are extracted by the CNN. Instead of using the last feature map of the CNN as the shared feature map, we propose to fuse the low-level feature maps with high-level feature maps. The RPN shares full-image convolutional feature maps with Fast R-CNN, and it outputs a set of rectangular object proposals with scores. At the same time, the position of each proposal is corrected by b-box regression, and some proposals with lower scores are deleted by non-maximum suppression (NMS). The proposals and shared feature maps are input into a region of interest (ROI) pooling layer to extract the higher-level features of each proposal. These features are fed into two fully connected layers, a box-regression layer and a box-classification layer, and finally the classification score and object bounding box can be obtained.

#### 3.2.1. Receptive Field

The receptive field is one of the most important concepts in CNN, which originated from biological neuroscience. The process of the convolution operation in CNNs is similar to that of biological neurons acting on organs [31]. The receptive field is defined as the region size of a pixel on the feature map mapped to the original image in the CNN. Since most feature extraction networks consist of convolution layer and pooling layer cascades, the receptive field is determined by the kernel size and stride of each layer. The calculation formula is as follows:(2)$$RF_{i}=RF_{i-1}+\left(Ksize_{i}-1\right)\prod_{i=1}^{n-1}stride_{i},$$
where $RF_{i}$ is the receptive field of the *i*-th layer, $RF_{i-1}$ is the receptive field of the (*i* − 1)-th layer, $stride_{i}$ is the stride of the *i*-th layer, and $Ksize_{i}$ is the kernel size of the *i*-th layer.

According to Formula (2), we calculated the receptive fields in different layers of VGG16, as shown in Table 2. It can be seen that the receptive fields of the high-level layers were larger than those of the low-level layers. The reason is that the pooling layers make the extracted feature maps smaller, and thus a pixel is mapped to a larger area of the original image. In object detection tasks, the attention of the detector is affected by the receptive field. In CNNs, if the receptive field is far larger than the scale of the object, it will produce a great deal of useless background noise. If the receptive field is far smaller than the scale of the object, it will not be able to extract the overall features of the object, affecting the experimental results [32]. According to this, we propose to select the proper layer for feature fusion according to the receptive field in Section 3.2.2.

#### 3.2.2. Feature Fusion Layer

In the feature extraction stage, VGG16 is adopted as the backbone, which contains five convolution blocks (Conv1, Conv2, Conv3, Conv4, Conv5). Max pooling with kernel size 2×2 and stride 2 is used after each convolution block. Therefore, the resolution of the feature is decreasing continuously. In addition, with the deepening of the network, the extracted feature becomes more abstract. In [33], the results of CNN visualization show that high-layer features contain more semantic information but less detail information, while low-layer features contain more detail information but suffer from the problem of background clutter and semantic ambiguity. The experiments in [27,34] demonstrated that the supplement of low-layer and high-layer features can improve the detection effect for multi-scale objects. The information gained is an especially important supplement for small objects, which require the higher spatial resolution provided by low-layer features. Inspired by them, we propose to select the appropriate layer for feature fusion, instead of just using the last feature map for detection. In Section 3.2.1, it is mentioned that for objects of different scales, the extracted features are affected by the receptive field. In [35], the feature extracted from the layer where the receptive field matches the object size is used to generate the region proposals for small objects. According to this idea, we believe that for small objects, when the receptive field matches the object size, more effective details can be extracted as a supplement to the high-level features.

By analyzing the scales of water surface floats in our dataset, we found that the scales of many floats were about 100×40 or smaller. So, according to the receptive field in Table 2, the low-level layer conv4_3 is quite proper. Therefore, we fuse the features of conv4_3 layer with the high-level layer conv5_3. The feature fusion module is shown in Figure 3. As the sizes of the feature maps in conv5_3 after the pooling layer are smaller than those of conv4_3, a deconvolutional operation is done to conduct upsampling in conv5_3. Then, this is followed by normalization with different scales (i.e., 10, 20). The Relu (Rectified Linear Unit) activation function is done before fusion. The fusion method we adopt here is the similar to that [27], which is a concatenation. The feature maps from different layers are concatenated along their channel axis. The size of the feature after fusion is unchanged, while only the number of channels is increased. Finally, the fusion feature maps are generated by a 1×1×512 convolutional layer, which can reduce dimension as well as feature combination. 

#### 3.2.3. Region Proposal Network

An RPN is a fully convolutional network which simultaneously predicts object bounding box and score at each position. The structure of the RPN is shown in Figure 4. The feature maps obtained by feature fusing are used as the input of the RPN, and the points of the feature maps correspond to the positions of the original image. A 3×3 sliding window is used to slide on the feature maps. Each sliding window is mapped to a lower-dimensional feature. The center of each sliding window corresponds to *k* anchors, and each anchor corresponds to a size and aspect ratio. The anchors are set in different scales and aspect ratios to cover the objects of different scales. However, in the original RPN, there is no method to determine the scales, aspect ratios and numbers of the anchors. If the scales of anchors do not match the size of the objects in experimental dataset, the detection effect will be affected. Hence, we propose to determine the scales, aspect ratios, and numbers of the anchors by analyzing the distribution of the objects in our dataset.

The distribution of the float scales in our dataset is shown in Figure 5. It can be seen that there were many small-scale floats and the width of the float was much larger than the height in our dataset. Therefore, according to the distribution, we assigned three aspect ratios {1:2, 2:5, 1:4} and four scales {32^2^, 64^2^, 128^2^, 256^2^} of the anchors, which were able to cover floats of different scales in our dataset. So, k=12 anchors will be produced at each sliding position in total. Then the box-regression layer will produce 4*k outputs, which are the coordinates of the *k* boxes. The box-classification layer will generate 2*k outputs, representing the probability that each box is an object or not. 

#### 3.2.4. Loss Function

Our network model has two output layers (box-classification layer and box-regression layer), which are both full-connection layers. Therefore, it follows multitask loss, and the loss function is defined as (3) [23]:(3)$$L\left(\left\{p_{i}\right\},\left\{t_{i}\right\}\right)=\frac{1}{N_{cls}}\underset{i}{\sum }L_{cls}\left(p_{i},p_{i}^{*}\right)+\lambda \frac{1}{N_{reg}}\sum_{i}p_{i}^{*}L_{reg}\left(t_{i},t_{i}^{*}\right).$$

The above formula was divided into two parts. The first part is the classification loss $L_{cls}$, and the second part is the regression loss $L_{reg}$. $L_{cls}\left(p_{i},p_{i}^{*}\right)$ is the logarithmic loss of two classes (object and not object), and $L_{reg}\left(t_{i},t_{i}^{*}\right)$ is the regression loss. They are shown in Equations (4) and (5):(4)$$L_{cls}\left(p_{i},p_{i}^{*}\right)=-\log\left[p_{i}^{*}p_{i}+\left(1-p_{i}^{*}\right)\left(1-p_{i}\right)\right]$$
(5)$$\;L_{reg}\left(t_{i},t_{i}^{*}\right)=R\left(t_{i}-t_{i}^{*}\right),\;\;$$
where *R* is the smooth *L*1 function, *i* is the index number of the anchor, and $p_{i}$ is the prediction probability of anchor *i* as the object.$\;p_{i}^{*}$ is the ground-truth label. $t_{i}=\left\{t_{x},t_{y},t_{w},t_{h}\right\}$ is a vector representing the four parametric coordinates of the predicted bounding box, and $t_{i}^{*}$ is the coordinate vector of the ground-truth box corresponding to the positive anchor. For bounding box regression, we adopt the parameterizations of the four co-ordinates as in Equations (6)–(9): [22]:(6)$$t_{x}=\frac{x-x_{a}}{w_{a}},\;t_{y}=\frac{y-y_{a}}{h_{a}},$$
(7)$$t_{w}=\log\left(\frac{w}{w_{a}}\right),\;t_{h}=\log\left(\frac{h}{h_{a}}\right),$$
(8)$$t_{x}^{*}=\frac{x^{*}-x_{a}}{w_{a}},\;t_{y}^{*}=\frac{h^{*}-y_{a}}{h_{a}},$$
(9)$$t_{w}=\log\left(\frac{w}{w_{a}}\right),\;t_{h}=\log\left(\frac{h}{h_{a}}\right),$$

### 3.3. Dataset Construction

Training network models based on deep learning require a large number of datasets. If the number of training samples is slightly less than the number of hyper-parameters, it will lead to over-fitting. There are no open datasets for water surface objects such as floats, so we constructed some training data manually and expanded the dataset by the data enhancement method. Data enhancement is obtains more data through operations such as rotation, cropping, color disturbance, and so on. The video surveillance stream for three days from the North Canal in Beijing was selected as our data source. We obtained 1065 images by video frame interception. In this paper, we adopted the methods of rotation and random clipping to expand the dataset, and finally obtained 2420 images, which were divided into training and testing data with the ratio of 7:3. For example, an original image is shown in Figure 6a, and the images after rotation and random clipping are shown in Figure 6b,c.

In accordance with the format of the VOC2007 dataset, we used the LabelImg tool to annotate floats in each image. The object Was surrounded by a rectangular bounding box and its category name was given. After annotating, the coordinate information of rectangular bounding box and the category information of object were saved in the generated XML file. Then, the txt files “train”, “trainval”, “test”, and “val” were generated according to the XML file.

### 3.4. Training

We adopted the method of alternating optimization to train our network model:Step 1:Training RPN. The RPN is initialized by the model parameters obtained on the ImageNet classification network. Then, end-to-end training is used for the RPN.Step 2:Training Fast R-CNN. The proposals obtained by Step 1 are adopted to perform end-to-end fine-tuned training of Fast R-CNN.Step 3:The RPN is finetuned by Fast R-CNN obtained in Step 2, while fixing the parameters of shared convolutional layers.Step 4:The region proposals obtained in Step 3 are used to fine-tune the fully connected layer of Fast R-CNN, while the shared convolutional layer is fixed.

## 4. Experiments

### 4.1. Setting

All of the experiments were implemented on a single NVIDIA GeForce GTX 1080Ti with 8 GB memory. The operating system was Ubuntu 16.04. The network model was implemented based on the popular and publicly available Caffe platform. We adopted the weight attenuation of 0.0005 and the momentum of 0.9. The training iterations were 50,000. We initialized the learning rate as 0.001 and it decreased 10 times every 20,000 iterations. The training time of the model was about 8 h.

### 4.2. Evaluation Criterion

#### 4.2.1. MAP

MAP (mean average precision) is an index to measure the recognition accuracy of the algorithm on object detection. The detections of different objects have their own curves according to the accuracy and recall rate. By integrating the curve function, the average precision (AP) of the area is obtained, and the average detection accuracy of all classes is the MAP. In this paper, because there is only one kind of object, the AP value of the algorithm is the same as the MAP value.

The precision represents the percentage of the number of objects correctly identified to the total number of objects identified. The recall ratio represents the percentage of objects correctly identified to the total number of objects. Their formulae are shown in Equations (10) and (11):(10)$$Precision=\frac{TP}{TP+FP},$$
(11)$$Recall=\frac{TP}{TP+FN},$$
where *TP* represents the number of samples correctly classified as positive (i.e., the number of samples that are actually positive and identified as positive by the model). *FP* represents the number of samples that are incorrectly classified as positive (i.e., the number of samples that are actually negative but identified as positive by the model). *FN* represents the number of instances that are incorrectly classified as negative (i.e., the number of samples that are actually positive but are identified as negative by the model).

#### 4.2.2. Detection Speed

Detection speed is another important performance index for object detection. In many applications, real-time detection is important. Frames per second (FPS), the number of pictures that can be detected by the algorithm in a second, is a common indicator for evaluating the detection speed. When comparing the FPS of different algorithms, object detection needs to be done under the same hardware conditions.

### 4.3. Analysis

#### 4.3.1. The Effect of Feature Fusion

We compared the detection results of feature fusion based on Faster R-CNN with non-feature fusion. In the experiment, the scales and aspect ratios of anchors were set as four scales {32^2^, 64^2^, 128^2^, 256^2^} with three ratios {1:2, 2:5, 1:4}. The precision–recall curves for the water surface float detection are shown in Figure 7, and the experimental results are shown in Table 3.

As can be seen from Table 3, the model with feature fusion had the better detection performance. In addition, feature fusion made the model more sensitive to small-scale objects, which will improve detection accuracy. The detection speed of Faster R-CNN with feature fusion was 11 FPS—slower than Faster R-CNN. The reason is that the extra feature fusion layer in the feature fusion model needs additional computation. However, our method still basically met the speed requirement of float detection. Figure 8a shows the detection result of Faster R-CNN with feature fusion, and Figure 8b shows the detection result of Faster R-CNN without feature fusion. It shows that the model with feature fusion could detect the water surface floats whether large or small. Therefore, it is important to fuse the high-level and low-level feature maps.

#### 4.3.2. Experiments on the Different Anchor Settings 

In Table 4 we investigate the detection of different anchor settings. It suggests that if just one anchor was generated at each position, the MAP dropped by 4–5%. The MAP was higher if using three scales with one aspect ratio or three aspect ratios with one scale. Using three scales {64^2^, 128^2^, 256^2^} with three ratios {1:2, 2:5, 1:4} was better than using three scales {128^2^, 256^2^, 512^2^} with three ratios {1:1, 1:2, 2:1} that were set in Faster R-CNN. This demonstrates that the detection effect were better if the scales and aspect ratios of anchors were close to the object scales in our dataset. Besides, using four scales with three ratios was better than using three scales with three ratios, suggesting that the detection effect can be improved by increasing the number of anchors.

#### 4.3.3. Comparison with Other Methods 

In order to validate the proposed method, we compared it with Fast R-CNN and YOLOv3 in the water surface float detection. The P–R curve of each model is drawn in Figure 9, and Table 5 shows the performance of the three methods. It suggests that the proposed method increased the detection accuracy by 8% compared with Fast R-CNN. Compared with YOLOv3, our proposed method increased the MAP from 78.6% to 83.7%. However, in terms of detection speed, YOLOv3 achieved a frame-rate of 35 FPS, which was faster than our proposed method. In the actual application scenario of this paper, the detection speed of ours could basically meet the requirements. 

In addition, we computed the recall at different IoU (Intersection over Union) ratios with ground truth. The Recall–IoU metric can be used to evaluate the localization performance, which is related to the final detection accuracy [36]. As shown in Figure 10, our proposed method obtained a comparable recall rate to YOLOv3 and Fast R-CNN. When the IoU was greater than 0.5, our proposed method achieved the best performance.

Several detection results of our proposed method and YOLOv3 are visualized in Figure 11. It can be seen that many small-scale floats were lost while using YOLOv3. While using the proposed method, floats with different shapes and scales in different scenes could be marked with high confidence.

## 5. Conclusions

In this paper, we consider a water surface float detection based on an improved Faster R-CNN in a natural scene. In response to the variance of sunlight, we use the gamma correction algorithm to preprocess the images. In addition, there are different scales of floats in our dataset, and the high-level feature maps only contain semantic features, which can lead to poor detection for small-scale objects. Hence, we fuse high-level features with low-level features, selecting the appropriate layers for feature fusion by analyzing the receptive fields of different layers. Moreover, we changed the scales and aspect ratios of anchors and increase the number of anchors, making them close to the scales of floats. The experimental results show that proposed method could achieve a MAP of 83.7% and a frame rate of 11 FPS on our dataset, which mostly meets the requirements of water surface float detection. This method can be applied to the automatic detection for water surface floats, which is meaningful and helpful to monitor river status.

However, our method still has some limitations. Firstly, existing datasets have a single category of floating objects, which makes them have poor generalization ability in engineering applications. We will collect various datasets, including common domestic garbage such as plastic bags, plastic bottles, and so on. Secondly, the gamma correction algorithm we adopted needs to determine the gamma value based on the dataset. Hence, we will study more effective image enhancement methods with stronger generalization ability. Thirdly, extra computation is generated because of feature fusion, which causes the detection speed to decrease. Therefore, the model needs to be further optimized, thus improving the detection speed. Furthermore, for particularly large-scale floats, Mask R-CNN [37] will be considered for semantic segmentation, thus quantifying the area of floats.

## Figures and Tables

**Figure 1 sensors-19-03523-f001:**
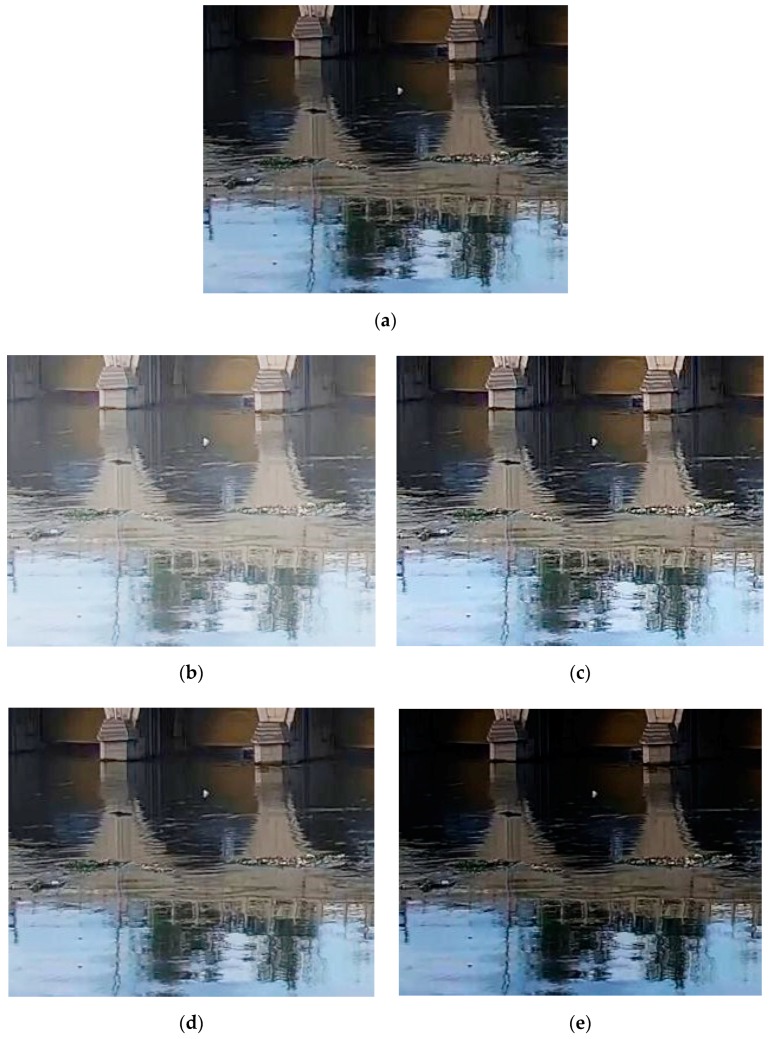
The corrected images with different gamma values. (**a**) Original image; (**b**) Corrected image when γ=0.4; (**c**) Corrected image when γ=0.6; (**d**) Corrected image when γ=0.8; (**e**) Corrected image when γ=1.3.

**Figure 2 sensors-19-03523-f002:**
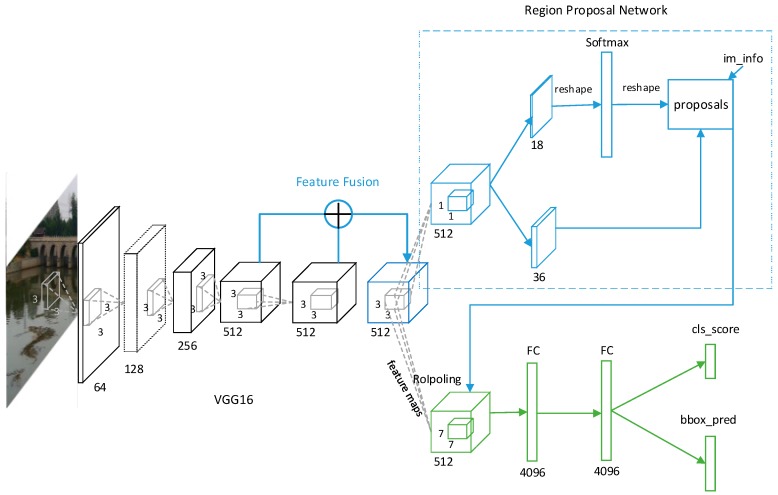
The scale-aware network based on Faster R-CNN.

**Figure 3 sensors-19-03523-f003:**
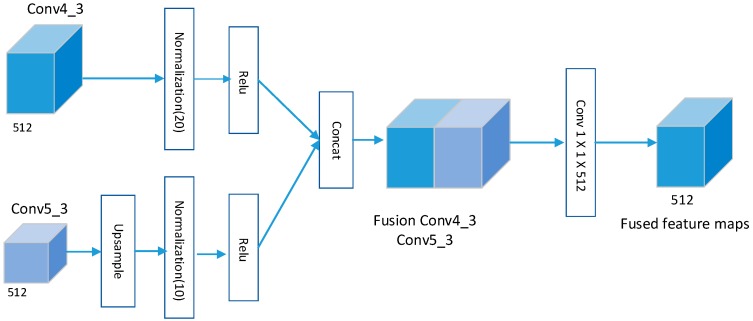
The feature fusion module.

**Figure 4 sensors-19-03523-f004:**
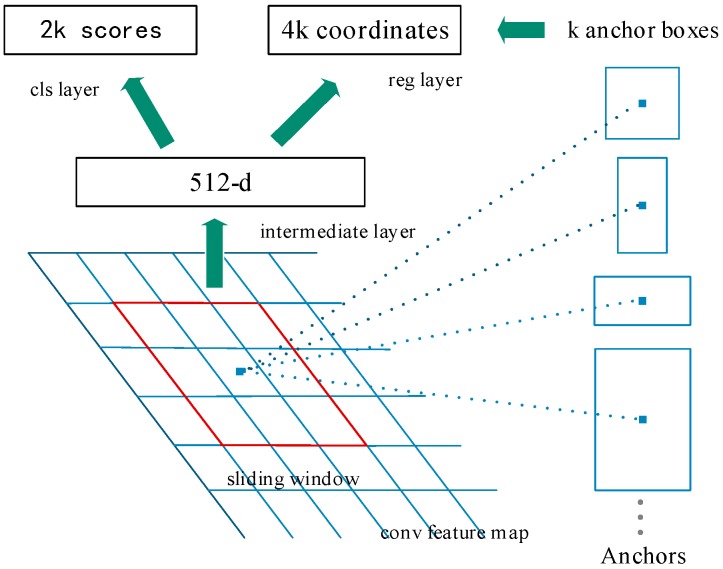
Structure of the region proposal network (RPN).

**Figure 5 sensors-19-03523-f005:**
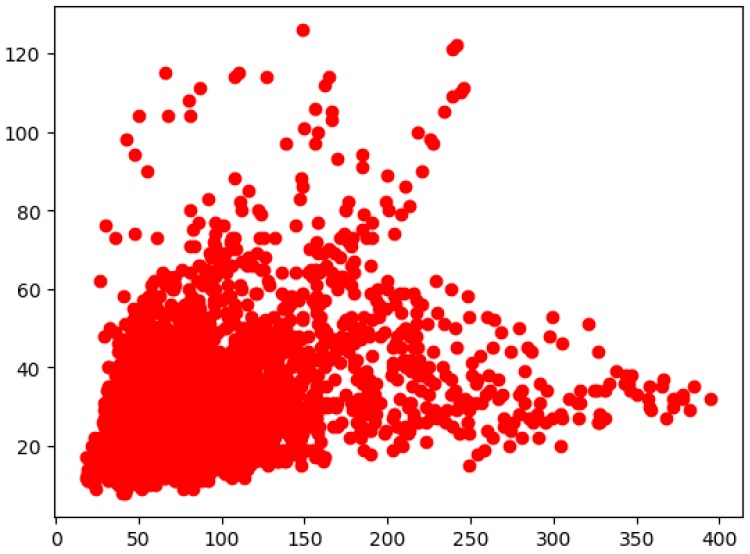
The distribution result of float scales.

**Figure 6 sensors-19-03523-f006:**
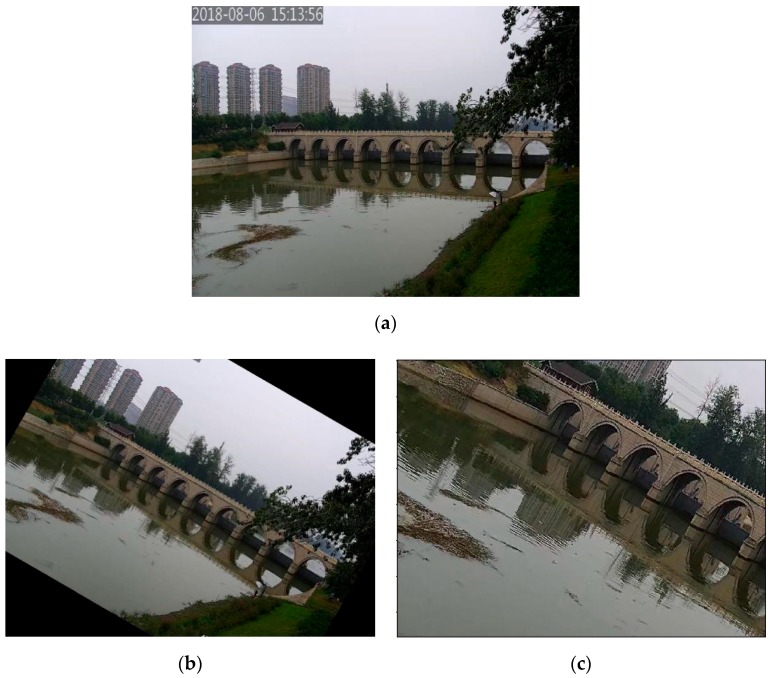
The results of data enhancement. (**a**) The original image; (**b**) The image after rotation; (**c**) The image after random clipping.

**Figure 7 sensors-19-03523-f007:**
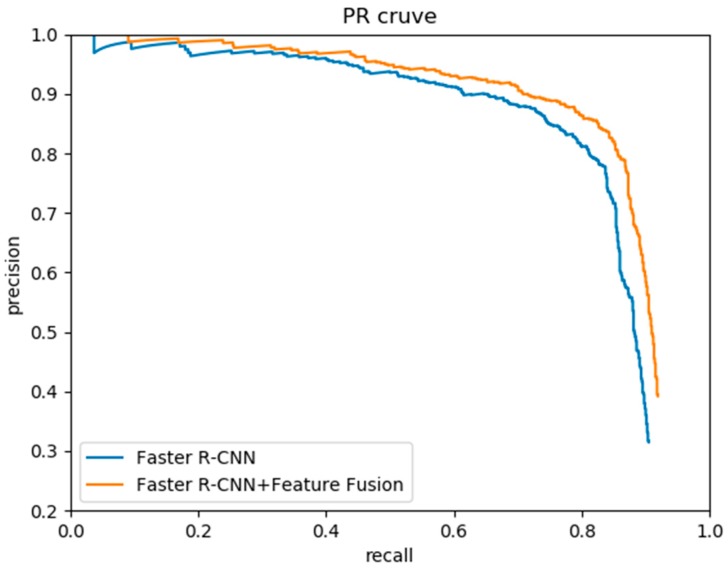
Precision–recall curves of Faster R-CNN with and without feature fusion.

**Figure 8 sensors-19-03523-f008:**
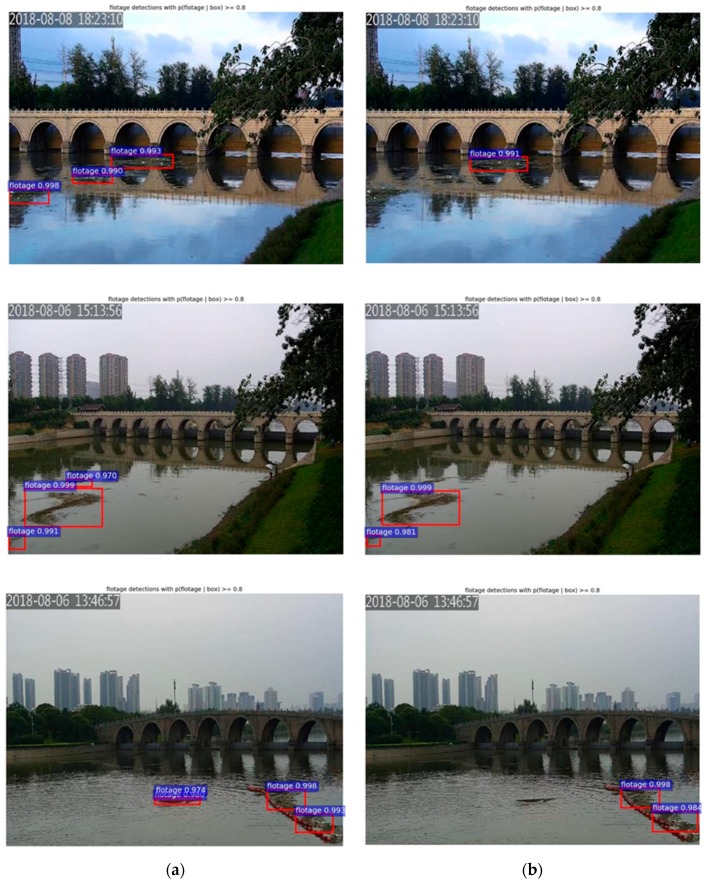
Detection results of Faster R-CNN (**a**) with and (**b**) without feature fusion.

**Figure 9 sensors-19-03523-f009:**
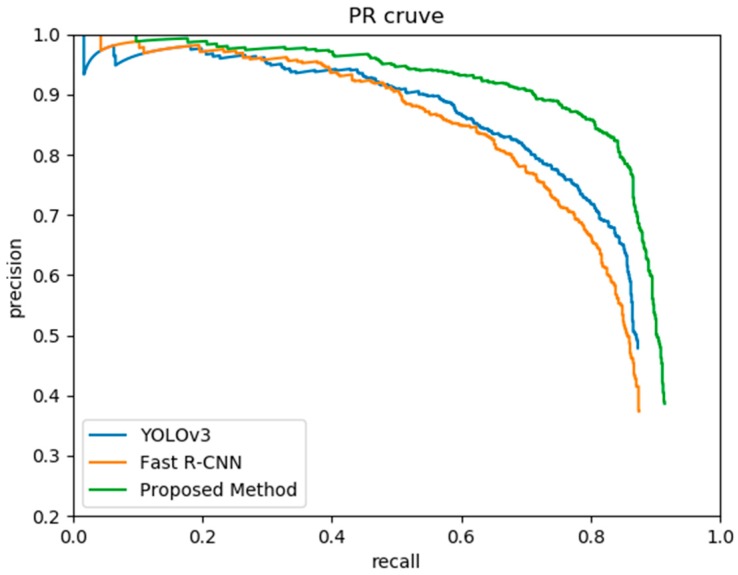
Precision–recall curves of YOLOv3, Fast R-CNN, and the proposed method.

**Figure 10 sensors-19-03523-f010:**
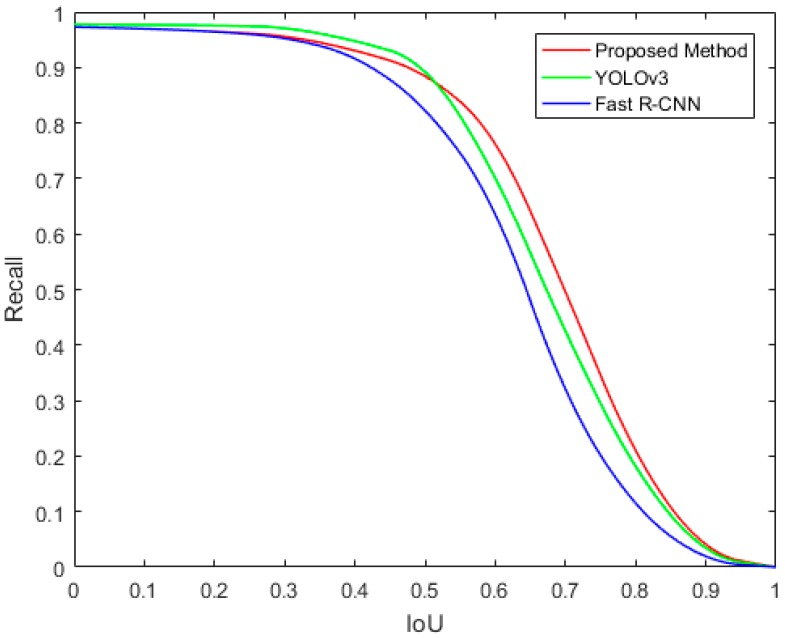
Recall–IoU curves of YOLOv3, Fast R-CNN, and the proposed method.

**Figure 11 sensors-19-03523-f011:**
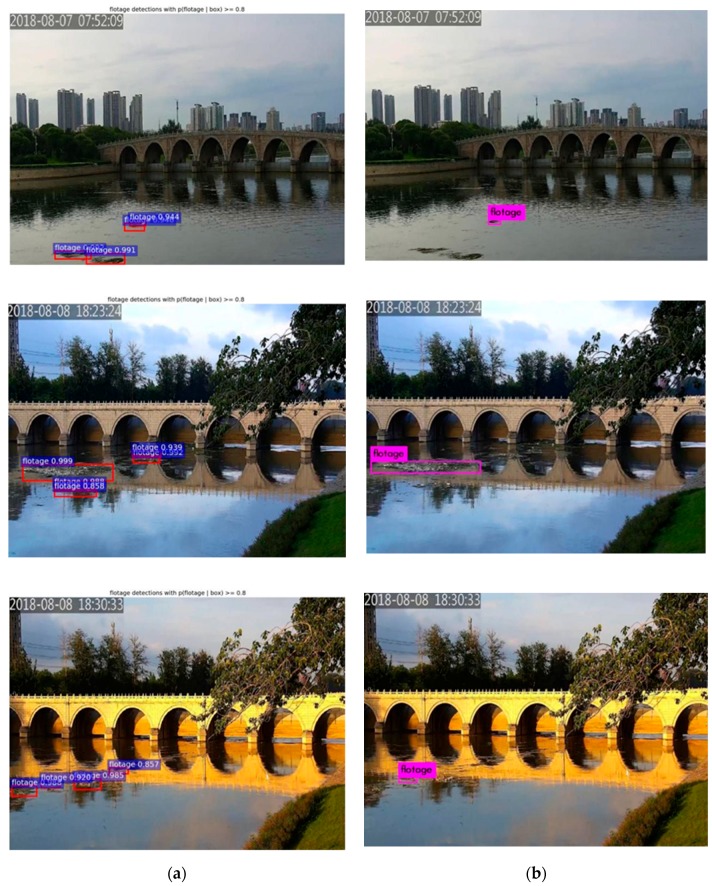
Comparison of the proposed method and YOLOv3. (**a**) The detection results using the proposed method; (**b**) The detection results using YOLOv3.

**Table 1 sensors-19-03523-t001:** The comparison of different gamma corrections.

Image	Original	γ=0.4	γ=0.6	γ=0.8	γ=1.3
Contrast	25,092	24,983	25,965	25,357	24,399

**Table 2 sensors-19-03523-t002:** The receptive fields (RFs) in different layers of VGG16.

Layer	RF
conv1_2	5
conv2_2	14
conv3_3	40
conv4_3	92
conv5_3	196

**Table 3 sensors-19-03523-t003:** The performance of Faster R-CNN with and without feature fusion. MAP: mean average precision.

Model	MAP	Speed (FPS)
Faster R-CNN	81.2%	13
Faster R-CNN + Feature Fusion	83.7%	11

**Table 4 sensors-19-03523-t004:** Detection results of different anchors.

Settings	Anchor Scales	Aspect Ratios	MAP (%)
1 scale, 1 ratio	64^2^	1:2	79.5%
	128^2^	1:2	78.9%
1 scale, 3 ratios	64^2^	{1:2, 2:5, 1:4}	81.8%
	128^2^	{1:2, 2:5, 1:4}	81.4%
4 scales, 1 ratio	{32^2^, 64^2^, 128^2^, 256^2^}	1:2	81.7%
	{32^2^, 64^2^, 128^2^, 256^2^}	2:5	82.0%
3 scales, 3 ratios	{128^2^, 256^2^, 512^2^}	{1:1, 1:2, 2:1}	81.2%
3 scales, 3 ratios	{64^2^, 128^2^, 256^2^}	{1:2, 2:5, 1:4}	82.6%
4 scales, 3 ratios	{32^2^,64^2^, 128^2^, 256^2^}	{1:2, 2:5, 1:4}	83.7%

**Table 5 sensors-19-03523-t005:** Experiments of the proposed method, YOLOv3, and Fast R-CNN.

Method	MAP (%)	Speed (FPS)
Fast R-CNN	75.3%	4
YOLOv3	78.6%	35
Proposed method	83.7%	11

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
