# Peer review of "Real-Time Water Surface Object Detection Based on Improved Faster R-CNN"

_sensors, 2019, doi:10.3390/s19163523_

Round 1
Reviewer 1 Report
The authors proposed a method to detect objects on the water surface using a Faster-RCNN algorithm that uses a fusion of characteristics based on information from receptive fields. However, the methodology adopted by the authors was not clear. In this sense, sections 3.2.1 and 3.2.2 should be reviewed.
The bibliographic review should be done again with the inclusion of more recent works. There were several citations throughout the text.
It is necessary to make clear the methodology adopted besides mentioning recent works that corroborate with the statements made by the authors. In addition, a review of English must be done, as I have located a few misspelled words and inadequate grammar.
In section 4.2 it is interesting to use other state-of-the-art metrics, such as the recall-IoU curve proposed by (Hosang et al., 2016). Another interesting metric would be the Mean Average Recall (mAR).
In results, it is important to highlight the possible causes of this decline in FPS. In addition, a better general discussion of results should be made. Why not use mask-RCNN? What are the limitations of the proposed algorithm?
What are the future works? Is the work completed? It is important to mention the Mask-RCNN.
Anyway, to be approved the article of a major review.

Author Response
Dear Reviewer,
Thank you very much for your constructive comments and suggestions on our manuscript entitled “Real-time Water Surface Object Detection Based on Improved Faster R-CNN”(sensors-556419). Those comments are all valuable and very helpful for revising and improving our paper, as well as the important guiding significance to our researches. We have studied comments carefully and have made correction which we hope meet with approval. Revised portion are marked in red in the paper. The point-by-point response to the comments are provided in the Word file.
Once again, thank you very much for your comments and suggestions。
Yours sincerely.
Yi Zhang

Reviewer 2 Report
The paper presents a method for detecting water surface objects in real time based on digital image processing and Faster-CNN algorithm. Authors prove by experiments that the proposed method is effective and has benefits with respect to other known methods like RPN. The documentation of the theme is well done, based on a diverse bibliography. The following observation must, however, be done:
1. All acronyms must be explained in the text. For example, B.P, KNN at line 76.
2. English must be deeply revised. There are many grammar mistakes. Some sentences must be reformulated, as they have no sense. Please ask a native English to supervise your text.
3. The elements that populate Table 1 are not explained and have no unit.
4. Section title 3.2.1. should be “Receptive field”.
Author Response
Dear Reviewer,
Thank you very much for the constructive comments and suggestions on our manuscript entitled “Real-time Water Surface Object Detection Based on Improved Faster R-CNN”(sensors-556419). Those comments are all valuable and very helpful for revising and improving our paper, as well as the important guiding significance to our researches. We have studied comments carefully and have made correction which we hope meet with approval. Revised portion are marked in red in the paper. The response to the comments are listed in the word.
Once again, thank you very much for your comments and suggestions。
Yours sincerely.
Yi Zhang

Reviewer 3 Report
Deep Learning is being used in almost all image related problems and in most cases it has been found to work better. The selected gamma value for image enhancement is based on a manual process and can be very subjective depending on the dataset. The goal is only to detect water movement that leads to object such as boat on the water. It is not clear how much tolerance is possible as water movement is not always tied to an object moving on the water, unless it is very calm water. As initial work, it is a good paper but requires different sets of data. For example, sea is rarely calm. Paper is not about object detection rather about water/float movement. I think title should reflect that. I would suggest authors to explain feature fusion more clearly. Authors have also not mentioned the time for training. Train/Test Split of 70:30 could have been 80:10:10, as train,validate and test.
Authors should also look at grammatical errors carefully or even getting it reviewed by an editor.
Author Response

(The authors gave the same response as above.)

Round 2
Reviewer 1 Report
In line 363, change the word "matric" by "metric".
Table 5 should be cited in the text.